# Nerve conduction, latency, and its association with hand function in young men

**Awad M. Almuklass**[1,2]*, **Abdulmajeed Mansour Alassaf**[1], **Rakan F. Alanazi**[1], **Turki Rashed Alnafisah**[1], **Thamir Ali Alrehaily**[1], **Yaser Al Malik**[1,2,3]

**1** Basic Medical Science Department, College of Medicine, King Saud Bin Abdulaziz University for Health Sciences, Riyadh, Saudi Arabia, **2** King Abdullah International Medical Research Center, Riyadh, Saudi Arabia, **3** Division of Neurology, King Abdulaziz Medical City, National Guard Health Affairs, Riyadh, Saudi Arabia

\* muklassa@ksau-hs.edu.sa

**Data Availability Statement:** All data underlying the findings described in the paper are available and uploaded as Supporting Information.

## Abstract

### Introduction

The median and ulnar nerves have been suggested to play a significant role in hand function; however, there are insufficient data to determine the strength of this association. This study aimed to investigate the correlation between hand function as measured with the Grooved pegboard test (GPT) and conduction velocity and latency of the median and ulnar nerves.

### Methods

We collected convenience samples in the College of Medicine, KSAU-HS. We used GPT to characterize hand function and performed measured nerve conduction velocity (NCV) and latency of the ulnar and median nerves of both hands. We used the Edinburgh handedness inventory (EHI) to determine hand dominance.

### Results

We recruited 28 healthy medical students aged 20–29 years (mean: 21.46 ± 1.62 years). Most were right-handed (n = 25, 89.3%), with a mean EHI score of 302 ± 210. The mean GPT time was significantly faster in the dominant (65.5 ± 6.4 s) than in the non-dominant (75.0 ± 9.6 s) hand. The NCV for the ulnar nerve of the dominant hand was significantly correlated with GPT (r = -0.52, p = 0.005) while median nerve was not correlated (0.24, p = 0.21). Regression analysis and collinearity test showed that the ulnar NCV explained 20% of the variance in GPT of the dominant hand ($R^2$ = 0.203, p = 0.016).

### Conclusion

The ulnar nerve conduction velocity, explained 20% of the variance in GPT times of the young men. Performance on this biomarker of neurological health seems to be more influenced by other factors in healthy young individuals.

**Funding:** The author(s) received no specific funding for this work.

**Competing interests:** The authors have declared that no competing interests exist.

## Introduction

Hands play a vital role in touch, communication, and performing activities of daily living. Hand function can be quantified with various tests of manual dexterity, such as the Grooved Pegboard Test (GPT) [1–3]. As a biomarker of neurological health across the lifespan, performance on pegboard tests reflects the ability of an individual to manipulate objects in a timely manner. Time to complete the GPT is predictive of motor performance capabilities in healthy adults 20–88 years [4] and is prolonged in individuals with neurological disorders [5, 6]. The test stresses aspects of cognitive acuity, tactile sensation, muscle strength, and force control [2, 4–8].

One potential explanatory variable for the variance in pegboard times that has received little attention is the integrity of peripheral nerves. This can be assessed with electrophysiological tests of nerve conduction velocity and the delay between an applied stimulus and an evoked action potential [9–11].

Neurological deficits can hinder mobility of the hand and the entire body, some of which can be diagnosed or treated using peripheral nerve investigations [9, 10].

In addition, sex may play a role in NCV and hand function. Females have a lower average NCV than males [12]; however, they perform better in overall hand function tests [12–15].

Numerous other factors influence the function of the peripheral nerves. One of the most important factors is nerve fiber diameter and myelination. Myelinated nerves fibers with larger diameters have faster NCV than unmyelinated nerve fibers with smaller diameters [16, 17]. Nerve length also plays a role in NCV. Shorter nerve fibers have a higher NCV than longer fibers [18].

In demyelinating disorders, myelin sheaths are damaged, resulting in deterioration of nerves, affecting the entire body, especially nerve conduction velocity and muscles resulting in worsening hand function [5, 11, 19]. Additionally, aging affects the nerves in the elderly, causing a slower nerve conduction velocity [13, 20, 21]. This decline is related to intrinsic changes in the muscle and neural interface resulting in weaker muscles and deteriorated hand function [7, 8, 12–15].

Although nerve conduction velocity and hand function, share many factors and are affected by similar diseases such as neurological disorders and aging, the relationship between hand function and nerve conduction velocity remains controversial. The purpose of study was to determine the association between nerve conduction velocity and latency of the median and ulnar nerves and time to complete the GPT in healthy young men. We hypothesized that nerves velocity and latency associate with hand function tested by GPT.

## Methods

### Study design, area, and settings

This was a cross-sectional single center study, carried out without follow-up. This study was conducted at the Physiology Laboratory of the College of Medicine, King Saud University of Health Sciences from November 2020 to September 2021. We recruited 31 healthy young male medical students aged 18–30 years, 3 of whom were excluded after screening for either a possible neuromuscular or skeletal disorder that might affect their performance on the GPT. We used a nonprobability, convenience sampling technique. All participants provided inform consent before participating.

### Data collection process

The ulnar and median nerves were studied using PowerLab technology. The Lafayette Instrument Grooved Pegboard Test was used to measure hand function for the dominant and non-

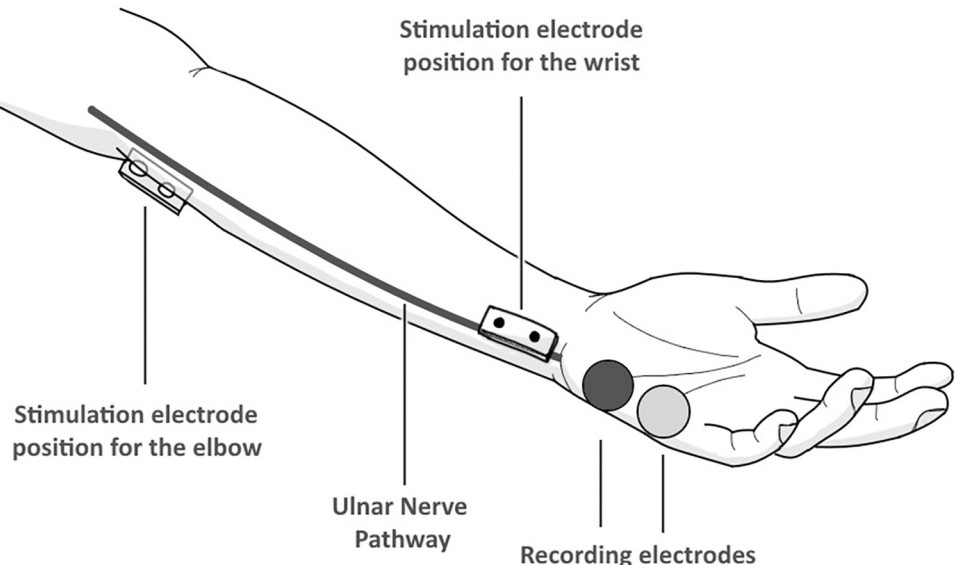

**Fig 1. Drawing that shows the pathway of the ulnar nerve and the positions of the electrodes.**

dominant hands. Three electrodes were placed on each nerve in each hand. To test the ulnar nerve, cathode and anode electrodes were placed on the hypothenar muscles in relation to the medial side of the palm below the little finger (Fig 1). The ground electrode was placed on the brachioradialis muscle relative to the lateral side of the forearm below the cubital fossa, electrode cream was placed on the stimulator before being placed under the elbow and on the medial side of the wrist to excite the ulnar nerve; both sites were marked to calculate the distance (Fig 1). Concerning the median nerve, the cathode and anode electrodes were placed on the thenar muscles, relative to the lateral side of the palm below the thumb (Fig 2). The ground electrode was placed on the superficial compartment of the forearm relative to the medial side

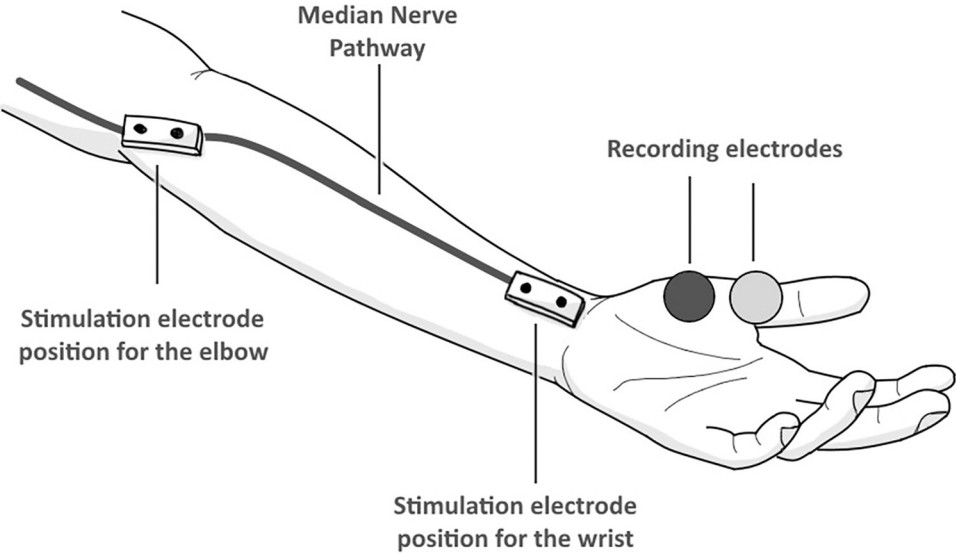

**Fig 2. Drawing that shows the pathway of the median nerve and the positions of the electrodes.**

of the forearm, electrode cream was placed on the stimulator before being placed near the cubital fossa and in the middle of the wrist; both sites were marked to calculate the distance. The electrical charge was set to 20 mA before energizing both nerves and recording NCV, wrist latency, and elbow latency. Concerning the Grooved Pegboard test, a stopwatch was used to keep track of the time taken by each hand to fill the holes as quickly as possible with the correct key position; the time was recorded.

The order of the tests (NCV and GPT) and the hand selection were counterbalanced for each participant. Other data collected included age and the Edinburgh Handedness Inventory, used to determine the handedness of each participant. Data were collected from paper and electronic devices. After screening, the participants read and signed an informed consent form (it included the research details, purpose of the research, and potential side effects or irritation during the examination) before taking the test. Data entry and recording were coded for each participant and only the investigators had access to the data. This study was approved by the Institutional Review Board of the King Abdullah International Medical Research Center, Ministry of National Guard-Health Affairs (approval number: SP20/385/R). All subjects were of legal age and were screened for health-related issues that might affect their hand function performance. Participants' identities were protected at all stages of the study. Randomly generated numbers were used to analyze the data. Moreover, all data collected were stored and protected using a password accessible only to the investigators.

## Data analysis

We used the statistical package for the social sciences (SPSS) software version 25 for data analysis. The Shapiro-Wilk test was first used to test for normality; this test was used to determine whether the datasets were normally distributed for each variable. Subsequently, an appropriate test was selected accordingly. Pearson's correlation was used to test for correlations between normally distributed variables, whereas Spearman's ρ was used for non-normally distributed variables. Regression analysis was used to develop a model that predicted hand function as measured by the GPT for the dominant, non-dominant, and combined hands. The collinearity test and variance inflation factor (VIF) were calculated to ensure that the regression estimates were stable and had minimal errors. The Wilcoxon signed-rank test was used to compare the performance of the dominant and non-dominant hands. To minimize type I errors in the analyses, the level of statistical significance was increased and set at $p < 0.025$. Cohen's $d$ effect size was calculated as the difference between the dominant and non-dominant hands.

## Results

We included 28 healthy men aged 20–29 years (mean ± SD age was 21.5 ± 1.6 years). Most of the participants were right handed (n = 25, 89.3%), with a mean Edinburgh handedness inventory score of 302 ± 210 (mean ± SD). The mean GPT was significantly ($p < 0.001$, Cohen's $d = 1.2$) faster in the dominant (65.5 ± 6.4 s) than in the non-dominant (74.96 ± 9.59s) hands (Table 1).

There were no significant differences between the Median and Ulnar nerve conduction velocities and latencies ($p > 0.05$, Cohen's $d$ range: 011–0.29). The mean Median NCV in the

**Table 1. Descriptive data of the population.**

| Age | EHI | GPT Dominant (s) | GPT Non-dominant (s) |
|---|---|---|---|
| 21.46 ± 1.62 | 301 ± 209 | 65.5 ± 6.4 | 75 ± 9.6 |

(mean ± SD). GPT, Grooved Pegboard Test. EHI, Edinburgh Handedness Inventory. (s): second

**Table 2. Descriptive data for GPT and nerve conduction studies and the correlation between dominant and non-dominant results.**

|  | GPT (s) | Ulnar velocity (m/s) | Ulnar Latency (s) | Median Velocity (m/s) | Median Latency (s) |
|---|---|---|---|---|---|
| **Dominant** | 65.5 ± 6.4 | 49.45 ± 8.63 | 0.00396 ± 0.00142 | 55.89 ± 7.56 | 0.00489 ± 0.000816 |
| **Non-dominant** | 75 ± 9.6 | 51.01 ± 7.52 | 0.00383 ± 0.00103 | 58.42 ± 10.02 | 0.0051 ± 0.00184 |
| **P value** | < 0.001 | 0.96 | 0.69 | 0.28 | 0.73 |
| Effect size (Cohen's *d*) | 1.2 | 0.19 | 0.11 | 0.29 | 0.14 |

(mean ± SD). GPT, Grooved Pegboard Test. (m/s): meter per second. (s): second

dominant hand was 55.886 ± 7.564 m/s and that in the non-dominant hand was 58.42 ± 10.02 m/s. The mean Median wrist latency in the dominant hand was 0.00489 ± 0.000816 s and that in the non-dominant hand was 0.0051 ± 0.00184 s (Table 2). The mean Ulnar NCV in the dominant hand was 49.446 ± 8.625 m/s and that in the non-dominant hand was 51.01 ± 7.52 m/s. The mean Ulnar wrist latency in the dominant hand was 0.00396 ± 0.00142 s and that in the non-dominant hand was 0.00383 ± 0.00103 s (Table 2).

When results from both hands were combined, the sample size was 56. The mean GPT score was 70.2 ± 9.4 s (Table 3). The Median NCV was 57.15 ± 8.89 m/s and the median latency was 0.004995 ± 0.001412 s. The Ulnar NCV was 50.23 ± 8.06 m/s and the ulnar latency was 0.003893 ± 0.001232 s (Table 3).

Correlation tests were performed after normality tests. Pearson's correlation was used for normal variables, whereas Spearman's rho was used for non-normal variables (Tables 4–6). Only the Ulnar NCV of the dominant hand was significantly correlated with the GPT (Pearson's r = -0.45, p = 0.016; Spearman's rho = -0.52, p = 0.005).

This indicates that the faster the Ulnar nerve conduction, the faster the performance of the GPT of the dominant hand. All variables (velocities and latencies for both nerves) were entered into a stepwise regression analysis. The collinearity test and variance inflation factor (VIF) were calculated to ensure that the regression estimates were stable and had minimal errors. The model revealed that Ulnar nerve conduction velocity of the dominant hand explained 20% of the variance in GPT of the dominant hand ($R^2$ = 0.203, p = 0.016, VIF = 1) (Fig 3).

## Discussion

This study investigated the associations between manual dexterity tested by the Grooved Pegboard test and nerve conduction velocity and latency of the median and ulnar nerves. Participants were recruited from a population of young men aged between 18 and 30 years to eliminate the possibility of confounding variables, since age and sex have been proven to influence nerve conduction [12, 14] and manual dexterity tested by GPT [8, 14, 15, 22]. The sample included right-handed (n = 25) and left-handed (n = 3) individuals, as a previous study showed that there is no difference between the results of the dominant right hand and the dominant left hand in the GPT score [22, 23]. Moreover, the order of the nerve study tests and the GPT for each hand was counterbalanced [22].

**Table 3. Descriptive data for the nerve conduction studies after combining the results from both hands of the sample.**

| cGPT (s) | cMNV (m/s) | cML (s) | cUNV (m/s) | cUL (s) |
|---|---|---|---|---|
| **70.2 ± 9.4** | 57.15 ± 8.89 | 0.004995 ± 0.001412 | 50.23 ± 8.06 | 0.003893 ± 0.001232 |

(mean ± SD). cGPT, Combined Grooved Pegboard test; cMNV, Combined Median Nerve Velocity; cML, Combined Median Latency; cUNV, Combined Ulnar Nerve Velocity; cUL, Combined Ulnar Latency; GPT, Grooved Pegboard Test. (m/s): meter per second. (s): second

**Table 4. Correlation between GPT and nerve studies results in the dominant hand.**

| Pearson Correlation | Age | dUL (s) | dUNV (m/s) | dML (s) | dMNV (m/s) |
|---|---|---|---|---|---|
| Dominant GPT (s) | -0.027 | -0.131 | -0.450* | -0.149 | -0.231 |
| P value | 0.892 | 0.506 | 0.016 | 0.451 | 0.237 |
| Spearman's correlation | Age | dUL (s) | dUNV (m/s) | dML (s) | dMNV (m/s) |
| Dominant GPT | -0.107 | -0.113 | -0.520* | -0.108 | -0.243 |
| P value | 0.588 | 0.568 | 0.005 | 0.583 | 0.212 |

dUL, Dominant Ulnar latency; dUNV, Dominant Ulnar Nerve Velocity; dML, Dominant Median Latency, Dmnv, Dominant Median Velocity; GPT, Grooved Pegboard Test.

* $p < 0.025$. (m/s): meter per second. (s): second

The GPT result was the only variable that showed a significant difference between the two hands, where the dominant hand had a faster result in most of the samples than the non-dominant hand. This is consistent with the literature [22, 23]. However, the other variables did not show any significant differences between the results of the two hands. This is consistent with previous studies suggesting that there are no differences in nerve conduction between the two hands [24]. These results suggest that dominance plays a role in manual dexterity but is not determined by the normal functioning of peripheral nerves, which suggests the involvement of upper motor neurons in the differences in both dominance and manual dexterity.

The only association observed in this study was between the GPT score and Ulnar NCV, which was only found in the dominant hand. In addition, regression analysis was used to develop a model that showed that 20.3% of the variance in the GPT score could be explained by Ulnar NCV. This is a novel finding that reveals few conditions. First, the ulnar nerve plays a bigger role in hand function than the median nerve, which could be explained by the nerve innervating certain muscles of the hand, such as the interosseous muscles, which is thought to be "the cornerstone of hand function" [25]. This is further supported by the fact that older subjects who have weakness of the abductor muscles and problems with pinching grip (both supplied by the ulnar nerve) have a slower GPT time [7, 8]. Second, these findings suggest that Ulnar NCV is more likely to be a limiting factor because the correlation was found only in the dominant hand. Hand dominance has a direct effect on manual dexterity, which improves how well a person can perform certain tasks tested in the GPT. With this improvement, the Ulnar NCV acts as a limitation that directly correlates with the GPT score. This possibility requires further research, as there is a lack of any additional data on this correlation to support the explanation; more data on older patients and patients with diseases that affect the ulnar nerve are needed.

**Table 5. Correlation between GPT and nerve studies results in the non-dominant hand.**

| Pearson Correlation | ndUL (s) | ndUNV (m/s) | ndML (s) | ndMNV (m/s) |
|---|---|---|---|---|
| Non dominant GPT (s) | 0.081 | -0.071 | 0.255 | 0.179 |
| P value | 0.683 | 0.718 | 0.19 | 0.361 |
| Spearman's correlation | NDUWL (s) | NDUNV (m/s) | NDMWL (s) | NDMNV (m/s) |
| Non dominant GPT (s) | 0.01 | 0.015 | 0.095 | 0.137 |
| P value | 0.961 | 0.939 | 0.632 | 0.486 |

ndUL, non-Dominant Ulnar latency; ndUNV, non-Dominant Ulnar Nerve Velocity; ndML, non-Dominant Median Latency; ndMNV, non-Dominant Median Velocity; GPT, Grooved Pegboard Test. (m/s): meter per second. (s): second

**Table 6. Correlation between GPT and nerve study results after combining the sample from both hands of the sample.**

| Pearson Correlation | cUL (s) | cUNV (m/s) | cML (s) | cMNV (m/s) |
|---|---|---|---|---|
| Combined GPT (s) | -0.045 | -0.144 | 0.175 | 0.109 |
| P value | 0.743 | 0.289 | 0.196 | 0.422 |
| Spearman's correlation | cUL (s) | cUNV (m/s) | cML (s) | cMNV (m/s) |
| Combined GPT (s) | -0.054 | -0.217 | -0.059 | 0.076 |
| P value | 0.69 | 0.108 | 0.664 | 0.577 |

cUL, Combined Ulnar Latency; cUNV, Combined Ulnar Nerve Velocity; cML, Combined Median Latency; cMNV, Combined Median Nerve Velocity. (m/s): meter per second. (s): second

This study had some limitations. First, the study sample was limited to healthy male medical students. This was done to reduce the confounding variables mentioned earlier. However, a larger study including female participants from the same cohort is required for comparison. Second, this study did not investigate the radial nerve effect on the GPT due to the lack of adequate equipment in our experimental setting. Future studies are needed to investigate this association in other populations from different cohorts, and to investigate the role of the radial nerve in hand function.

## Conclusion

There was a significant difference in the hand function tested by GPT between the dominant and non-dominant hands, which was not present with nerve conduction velocities and latency of the Ulnar and Median nerves. The Ulnar NCV in the dominant hand was the only variable

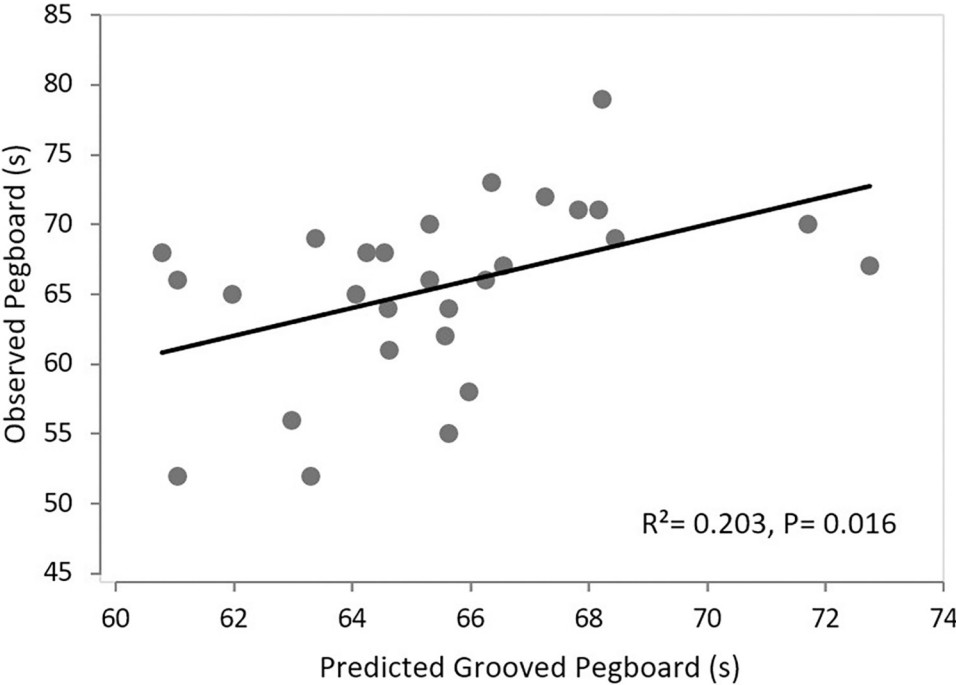

**Fig 3. The constructed graph using the produced regression formula for the observed and predicted Grooved Pegboard performance time measured in second.** The Ulnar NCV in the dominant hand explains 20% of the variance in GPT.

that correlated with GPT and explains 20% of the variance. This suggests that other variables that play larger roles in manual dexterity and dominance should be investigated.

## Supporting information

**S1 Data.**
(XLSX)

## Author Contributions

**Conceptualization:** Awad M. Almuklass, Thamir Ali Alrehaily.

**Data curation:** Awad M. Almuklass, Turki Rashed Alnafisah, Yaser Al Malik.

**Formal analysis:** Awad M. Almuklass, Abdulmajeed Mansour Alassaf, Yaser Al Malik.

**Investigation:** Abdulmajeed Mansour Alassaf, Rakan F. Alanazi, Turki Rashed Alnafisah, Thamir Ali Alrehaily.

**Methodology:** Awad M. Almuklass, Abdulmajeed Mansour Alassaf, Rakan F. Alanazi, Turki Rashed Alnafisah, Thamir Ali Alrehaily.

**Project administration:** Abdulmajeed Mansour Alassaf.

**Supervision:** Awad M. Almuklass, Yaser Al Malik.

**Validation:** Awad M. Almuklass, Rakan F. Alanazi, Yaser Al Malik.

**Visualization:** Awad M. Almuklass, Turki Rashed Alnafisah, Thamir Ali Alrehaily.

**Writing – original draft:** Awad M. Almuklass.

**Writing – review & editing:** Awad M. Almuklass, Abdulmajeed Mansour Alassaf, Rakan F. Alanazi, Turki Rashed Alnafisah, Thamir Ali Alrehaily, Yaser Al Malik.

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
