## [Decision Letter · Decision Letter 0]

5 Aug 2024

PONE-D-23-26799Nerve Conduction and Latency and its Association with Hand Function in Young MenPLOS ONE

Dear Dr. Almuklass,

Thank you for submitting your manuscript to PLOS ONE. After careful consideration, we feel that it has merit but does not fully meet PLOS ONE’s publication criteria as it currently stands. Therefore, we invite you to submit a revised version of the manuscript that addresses the points raised during the review process.

We look forward to receiving your revised manuscript.

Kind regards,

Engy Asem Ashaat

Academic Editor

PLOS ONE

Journal Requirements:

3. In this instance it seems there may be acceptable restrictions in place that prevent the public sharing of your minimal data. However, in line with our goal of ensuring long-term data availability to all interested researchers, PLOS’ Data Policy states that authors cannot be the sole named individuals responsible for ensuring data access (http://journals.plos.org/plosone/s/data-availability#loc-acceptable-data-sharing-methods).

Reviewers' comments:

Reviewer's Responses to Questions

**Comments to the Author**

1. Is the manuscript technically sound, and do the data support the conclusions?

Reviewer #1: Partly

2. Has the statistical analysis been performed appropriately and rigorously? 

Reviewer #1: I Don't Know

3. Have the authors made all data underlying the findings in their manuscript fully available?

Reviewer #1: Yes

4. Is the manuscript presented in an intelligible fashion and written in standard English?

Reviewer #1: Yes

5. Review Comments to the Author

Reviewer #1: There are minor changes is needed that I have inserted my comments in my recommendations document like

Some comments in abstract , introduction , tables and discussion. There is rearrangement of some paragraphs in introduction and explanation to certain abbreviations.

6. PLOS authors have the option to publish the peer review history of their article (what does this mean?). If published, this will include your full peer review and any attached files.

Reviewer #1: No

---

## [Author Response · Author response to Decision Letter 0]

28 Aug 2024

Response to reviewers:

 Authors thank the reviewers for the comments and suggestions

Reviewer comment: “Delete as it is repeated in the same sentence.”

Done.

Reviewer comment: “Why you do not give any hint about median nerve here?” 

Done.

Reviewer comment: “Omit dot.”

Done.

Reviewer comment: “Move this paragraph after paragraph from 116-119.”

Done.

Reviewer comment: “Add the meaning for these abbreviations below the table.”

As recommended, the meaning for these abbreviations were written under all tables of need.

Dear Solna Carreon Santos from PLOS ONE

Greetings,

Kindly find my responses to your comments on the submitted revision.

“1. Your ethics statement should only appear in the Methods section of your manuscript. If your ethics statement is written in any section besides the Methods, please delete it from any other section.”

The statement was deleted since it is written in the methods section.

“2. In the online submission form, you indicated that [The data used to generate the findings of this study are available from the corresponding author upon reasonable request.].

This policy applies to all data except where public deposition would breach compliance with the protocol approved by your research ethics board. If your data cannot be made publicly available for ethical or legal reasons (e.g., public availability would compromise patient privacy), please explain your reasons on resubmission and your exemption request will be escalated for approval.”

All data underlying the findings described in the manuscript are available and uploaded as supplementary information.

“3. Please upload a copy of Figures 1, 2 and 3 which you refer to in your text on pages 8 an 12. Or if the figure is no longer to be included as part of the submission please remove all reference to it within the text.”

Figure 1, 2 and 3 were uploaded in the text in the right pages.

Kindest Regards

Awad M. Almuklass, PhD

---

## [Editor Report · Decision Letter 1]

8 Sep 2024

Nerve Conduction and Latency and its Association with Hand Function in Young Men

PONE-D-23-26799R1

Dear Dr. Almuklass,

We’re pleased to inform you that your manuscript has been judged scientifically suitable for publication and will be formally accepted for publication once it meets all outstanding technical requirements.

Kind regards,

Engy Asem Ashaat

Academic Editor

PLOS ONE
---

## [Editor Report · Acceptance letter]

19 Sep 2024

PONE-D-23-26799R1 

PLOS ONE

Dear Dr. Almuklass, 

I'm pleased to inform you that your manuscript has been deemed suitable for publication in PLOS ONE. Congratulations! Your manuscript is now being handed over to our production team.

Kind regards, 

on behalf of

Professor Engy Asem Ashaat 

Academic Editor

PLOS ONE